# Heat stress impairs centromere structure and segregation of meiotic chromosomes in *Arabidopsis*

**Lucie Crhak Khaitova[1], Pavlina Mikulkova[1], Jana Pecinkova[1], Manikandan Kalidass[2], Stefan Heckmann[2], Inna Lermontova[2], Karel Riha[1]***

[1]CEITEC Masaryk University, Brno, Czech Republic; [2]Leibniz Institute of Plant Genetics and Crop Plant Research (IPK) Gatersleben, Gatersleben, Germany

**Abstract** Heat stress is a major threat to global crop production, and understanding its impact on plant fertility is crucial for developing climate-resilient crops. Despite the known negative effects of heat stress on plant reproduction, the underlying molecular mechanisms remain poorly understood. Here, we investigated the impact of elevated temperature on centromere structure and chromosome segregation during meiosis in *Arabidopsis thaliana*. Consistent with previous studies, heat stress leads to a decline in fertility and micronuclei formation in pollen mother cells. Our results reveal that elevated temperature causes a decrease in the amount of centromeric histone and the kinetochore protein BMF1 at meiotic centromeres with increasing temperature. Furthermore, we show that heat stress increases the duration of meiotic divisions and prolongs the activity of the spindle assembly checkpoint during meiosis I, indicating an impaired efficiency of the kinetochore attachments to spindle microtubules. Our analysis of mutants with reduced levels of centromeric histone suggests that weakened centromeres sensitize plants to elevated temperature, resulting in meiotic defects and reduced fertility even at moderate temperatures. These results indicate that the structure and functionality of meiotic centromeres in *Arabidopsis* are highly sensitive to heat stress, and suggest that centromeres and kinetochores may represent a critical bottleneck in plant adaptation to increasing temperatures.

## eLife assessment

This study is an **important** contribution to our insights into the impact of heat stress on sexual reproduction in plants and provides information about how centromere integrity is affected by heat stress during male meiosis in *Arabidopsis thaliana*. The evidence supporting the claims, specifically the dynamics of tagged proteins in meiocytes by live cell imaging is **solid**, even though a deeper mechanistic understanding is still lacking.

## Introduction

Global warming impacts crop productivity with more frequent and extreme heatwaves being particularly damaging (*Brás et al., 2021*). The lower yields associated with heatwaves are mainly attributed to the impairment of the plant reproductive system (*Lippmann et al., 2019*). Thus, a fundamental understanding of how heat stress affects plant reproduction can guide breeding strategies toward generating climate change-resilient crops. Heat stress impairs both male and female reproductive structures, but male gametogenesis is particularly sensitive to higher temperatures (*Zinn et al., 2010*). Studies in crop and non-crop species have shown that heat stress affects pollen count, morphology,

**\*For correspondence:**
karel.riha@ceitec.muni.cz

**Competing interest:** The authors declare that no competing interests exist.

viability, dehiscence, germination, and pollen tube growth (*De Storme and Geelen, 2014*; *Chaturvedi et al., 2021*).

Acute heat stress applied at different stages of flower development has been found to have the most detrimental effect during microsporogenesis (*Pécrix et al., 2011*; *Hedhly et al., 2020*). Meiosis, a reductive cell division that halves number of chromosomes in two consecutive rounds of chromosome segregation, is a crucial step of microsporogenesis. It is a relatively slow process, lasting approximately 1.5–2 days in *Arabidopsis* and barley (*Armstrong et al., 2003*; *Higgins et al., 2012*; *Prusicki et al., 2019*; *Valuchova et al., 2020*). Prophase I, the longest phase of meiosis, is characterized by pairing and recombination of homologous chromosomes to form stable bivalents. It is followed by two chromosome segregation cycles that first divide homologous chromosomes and then the sister chromatids. Heat stress affects various aspects of plant meiosis. In prophase I, elevated temperature alters the rate and distribution of meiotic recombination (*Higgins et al., 2012*; *Lloyd et al., 2018*; *Modliszewski et al., 2018*) and a recent study in *Arabidopsis* has shown that heat shock response pathway directly regulates the recombination machinery (*Kim et al., 2022*). More severe heat stress can impair synapsis and pairing of homologous chromosomes (*Loidl, 1989*; *De Storme and Geelen, 2020*; *Ning et al., 2021*). This is likely due to inefficient completion of homologous recombination, which is monitored by specialized pachytene checkpoint (*De Jaeger-Braet et al., 2022*).

Heat stress affects the later stages of meiosis as well. Elevated temperatures disturb spindle orientation during meiotic divisions and the formation of radial microtubule arrays, resulting in aberrant cytokinesis and diploid microspores (*Pécrix et al., 2011*; *De Storme and Geelen, 2020*). Furthermore, acute heat stress leads to chromosome mis-segregation and the formation of micronuclei (*Wang et al., 2017*; *De Storme and Geelen, 2020*; *De Jaeger-Braet et al., 2022*). The meiotic micronuclei and chromosomal aberrations may arise as a consequence of abnormal repair of meiotic DNA breaks or homologous recombination intermediates in prophase I. However, they can also be caused by defects in chromosome segregation during meiotic divisions (*Fenech et al., 2011*). Proper attachment of kinetochores to the spindle microtubules before anaphase is crucial for faithful chromosome partitioning to daughter cells, and this process is controlled by spindle assembly checkpoint (SAC). Therefore, micronuclei typically occur in mutants with impaired centromere/kinetochore structure or SAC when treated with spindle inhibitors (*Kalitsis et al., 2000*; *Lermontova et al., 2011a*).

In most eukaryotes centromeres are confined to a restricted region on each chromosome that serves as a platform for assembly of the kinetochore, a multiprotein structure that connects chromosome with the spindle. Centromeres are defined epigenetically by a specialized centromeric histone H3 variant (CENH3), which forms the foundation for the recruitment of kinetochore proteins (*McAinsh and Marston, 2022*). In contrast to canonical histones, replicative dilution of CENH3 is not replenished immediately in S-phase, but in subsequent stages of the cell cycle (*Stirpe and Heun, 2023*). In plants, CENH3 is loaded on centromeres in G2 and persists there throughout the cell cycle (*Talbert et al., 2002*; *Lermontova et al., 2006*). CENH3 is a very stable component of centromeric chromatin, although its amount at centromeres gradually declines in terminally differentiated cells in plants and animals (*Lermontova et al., 2011b*; *Swartz et al., 2019*).

In this study, we have discovered that heat stress significantly reduces the amount of CENH3 on meiotic chromosomes in *Arabidopsis thaliana*. This loss of CENH3 leads to the formation of micronuclei, which in turn contributes to the decrease in pollen formation and fertility in plants exposed to heat stress. Additionally, we have found that plants with a genetic mutation that reduces the amount of centromeric histone are more sensitive to moderately elevated temperature. These results suggest that meiotic centromeres may represent a crucial point of vulnerability for plants in adaptation to raising temperatures.

## Results

### Pollen production and fertility decline with increasing temperature

In our previous work, we identified the *cenh3-4* allele of *Arabidopsis* centromeric histone *CENH3* that carries a mutation in the splicing donor site of the third exon (*Capitao et al., 2021*). This leads to a 10-fold reduction in fully spliced *CENH3* mRNA and a decreased amount of centromeric histone. Consequently, *cenh3-4* plants have smaller centromeres and are sensitive to oryzalin (*Capitao et al., 2021*). Under standard conditions, *cenh3-4* mutants are barely distinguishable from wild type, but

we noticed their reduced fertility when grown at an elevated temperature. To systematically assess this phenotype, we grew plants under standard conditions (21°C) until they formed four true leaves, and then continued their cultivation in growth chambers tempered to 16°C, 21°C, 26°C, and 30°C (*Figure 1—figure supplement 1*). At 16°C, plants exhibited the slowest growth, but also the highest fertility, as assessed by pollen count and silique length, which is indicative of seed yield (*Figure 1*). Fertility slightly decreased at 21°C and further declined at 26°C. In *cenh3-4* mutants, we noticed a sharp decline in pollen production and fertility at 26°C (86±48 pollen per anther), whereas the fertility of the wild type was still relatively high (213±58 pollen per anther) (*Figure 1B and C*). Both *cenh3-4* and wild type plants became infertile at 30°C. The heat-induced sterility was reversible and *cenh3-4* as well as wild type plants transferred from 30°C to 21°C regained fertile flowers (*Figure 1—figure supplement 1C*). These data indicate that pollen production and fertility gradually decline with increasing temperature and this trend is particularly pronounced in *cenh3-4* mutants that have become almost sterile already at 26°C.

It has been reported that extreme heat stress alters chromosome segregation fidelity and the duration of *Arabidopsis* meiosis (*De Jaeger-Braet et al., 2022*). Temperatures of 34°C and above abolished chromosome pairing and synapsis and led to the formation of univalents. However, meiotic chromosomes are fully paired at 30°C (*Ning et al., 2021*; *De Jaeger-Braet et al., 2022*; *Fu et al., 2022*), indicating that the recombination defects are not primarily responsible for abortive pollen development at this temperature.

## Heat stress delays meiotic progression and induces micronuclei

To assess the effect of temperature on meiotic progression and chromosome segregation at temperatures up to 30°C, we performed live imaging of meiotic divisions in the HTA10:RFP reporter line, which marks chromatin (*Valuchova et al., 2020*). We measured the duration of meiotic divisions from the end of diakinesis until the formation of haploid nuclei in telophase II (*Figure 2—figure supplement 1*). Meiotic divisions were slowest at 16°C and lasted, on average, 441 and 459 min in wild type and *cenh3-4*, respectively (*Videos 1 and 2*, *Figure 2A*). Meiosis progressed significantly faster with increasing temperature. Wild type meiotic divisions took 169 min at 21°C and 142 min at 26°C (*Videos 3 and 4*). Meiotic divisions were also rapid in *cenh3-4* at 21°C, lasting on average 156 min, but slowed down to 238 min at 26°C (*Videos 5 and 6*). This contrasts with the situation in wild type, where divisions occur at the fastest rate at 26°C. Interestingly, meiotic divisions were prolonged at 30°C in both wild type and *cenh3-4* mutants to 286 and 264 min, respectively (*Videos 7 and 8*). The slowdown of meiosis at 26°C and 30°C in *cenh3-4*, and at 30°C in wild type coincides with the steep decline of pollen production (*Figure 1B and C*) and indicates problems with meiotic progression.

Live imaging in *cenh3-4* plants and, at 30°C, also in wild type showed formation of micronuclei that began to form during meiosis I and persisted till telophase II (*Videos 7 and 8*). We quantified the micronuclei in fixed anthers at the tetrad stage using confocal microscopy (*Figure 2B and C*). The micronuclei were apparent in *cenh3-4* mutants at all temperatures, which is consistent with partially impaired centromere function in these plants (*Capitao et al., 2021*). Nevertheless, their occurrence substantially increased at 26°C and severe defects in chromosome segregation were observed at 30°C (*Figure 2B and C*; *Videos 6 and 8*) resulting in very few regular tetrads. Whereas we occasionally detected micronuclei at lower temperatures also in wild type, albeit at a much lower level than in *cenh3-4*, their incidence increased 17-fold between 26°C and 30°C (*Figure 2B and C*, *Videos 4 and 7*). In addition, we observed dyads and polyads, which is consistent with the previous study performed at a similar temperature (*De Storme and Geelen, 2020*). These data show that temperature-induced fertility reduction coincides with increased occurrence of micronuclei.

Micronuclei may arise from acentric fragments and dicentric chromosomes derived from defective repair of meiotic breaks. Indeed, temperatures above 30°C were reported to cause aberrant recombination intermediates and induce pachytene checkpoint (*De Storme and Geelen, 2020*; *De Jaeger-Braet et al., 2022*). In this scenario, the absence of meiotic breaks in SPO11-deficient plants should prevent the formation of temperature-induced micronuclei. *Arabidopsis spo11-2* mutants do not form bivalents, and homologous chromosomes segregate randomly in anaphase I (*Stacey et al., 2006*; *Hartung et al., 2007*). Cytogenetic analysis in *spo11-2* plants revealed a relatively high level of micronuclei in tetrads at 21°C (*Figure 2B and C*). Because unpaired univalents cannot properly biorient at metaphase I spindle, these micronuclei are likely generated by chromosome mis-segregation.

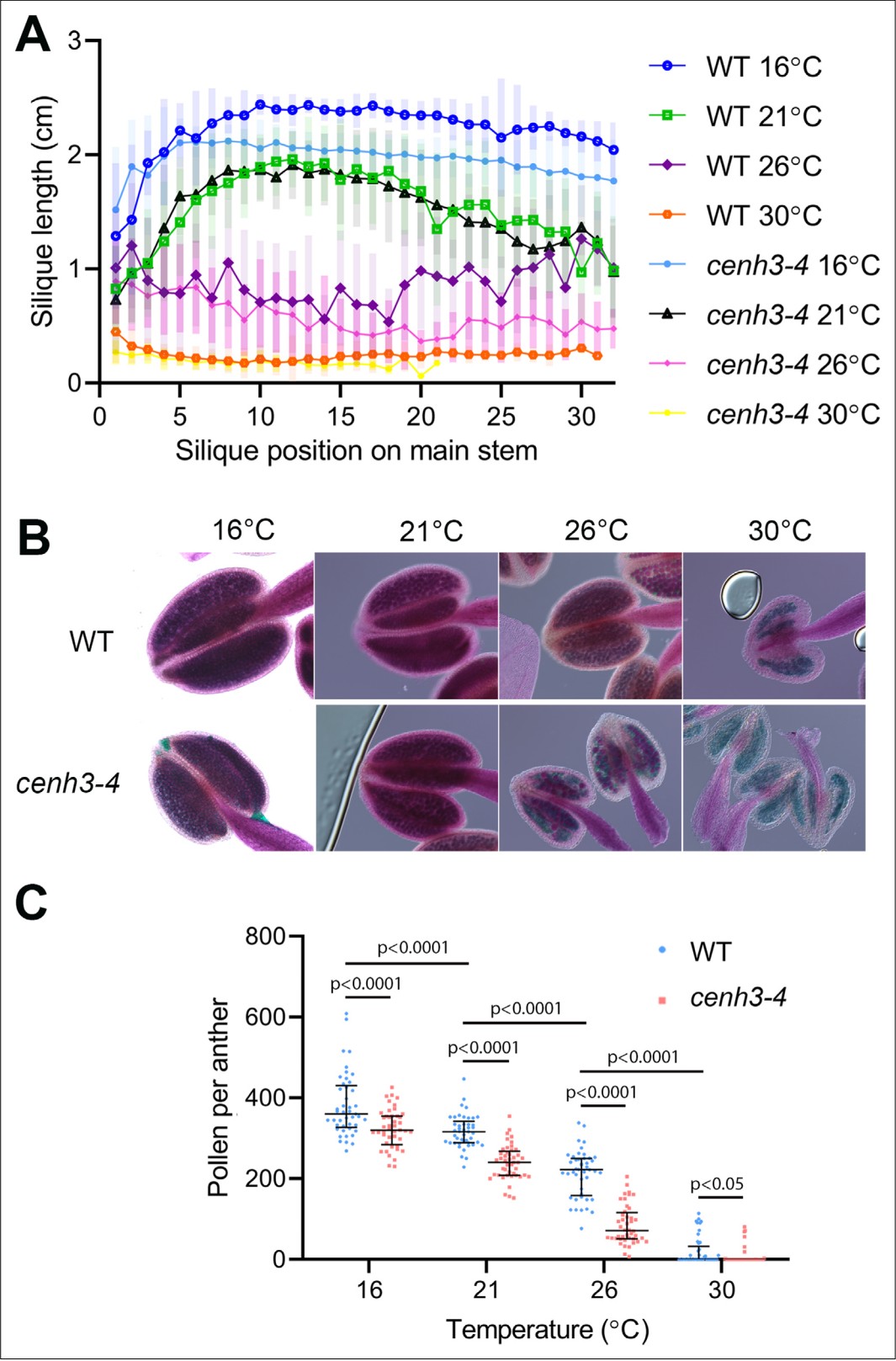

**Figure 1.** Effect of temperature on fertility in wild type and *cenh3-4* plants. (**A**) Analysis of silique length through the main stem of wild type (WT) grown at 16°C (n=18), 21°C (n=25), 26°C (n=13), 30°C (n=20) and *cenh3-4* mutant at 16°C (n=22), 21°C (n=20), 26°C (n=22), and 30°C (n=11). The silique position is numbered from the oldest to the youngest silique on the main stem. Error bars depict standard deviation. (**B**) Anthers of the abovementioned plants

*Figure 1 continued on next page*

*Figure 1 continued*

after Alexander staining. (**C**) Quantification of viable pollen per anther (n=45). Significance of the difference is counted using two-tailed t-test. Source values for (**A**) and (**C**) are available in *Source data 1*.

The online version of this article includes the following figure supplement(s) for figure 1:

**Figure supplement 1.** Effect of temperature on growth of wild type and *cenh3-4* plants.

Importantly, number of the micronuclei almost doubled in *spo11-2* plants grown at 30°C (*Figure 2B and C*) arguing that temperature has a direct effect on chromosome segregation.

In its natural habitats, *A. thaliana* usually flowers from March to early summer under average daily temperatures lower than the ones used for cultivation in laboratories (*Shindo et al., 2007*; *Brachi et al., 2010*). Due to the day-night cycle, they never experience sustained temperatures over 30°C. Therefore, we tested whether meiotic defects observed after continuous cultivation at 30°C could also be induced under more physiological conditions that mimic a hot day. To this end, we cultivated plants in chambers tempered to 18°C overnight and increased temperature to 34°C for 6 hr during the day (*Figure 2—figure supplement 2*). We observed a drastic reduction in fertility and pollen count, as well as an increased frequency of micronuclei compared to control plants grown under the 18°C/21°C night/day regime (*Figure 2—figure supplement 2B–D*). These data suggest that heatwaves occurring during flowering can have a detrimental effect on *Arabidopsis* meiosis.

## Heat stress reduces the amounts of centromeric histone on meiotic centromeres

Our phenotypic analysis indicates that *cenh3-4* mutants grown at 26°C exhibit the same behavior as wild type plants grown at 30°C. Therefore, we investigated whether increasing temperature weakens the centromere structure, which could explain the temperature sensitivity of *cenh3-4* plants with less centromeric histone. CENH3 is present on meiotic chromosomes from early prophase I (*Talbert et al., 2002*; *Lermontova*

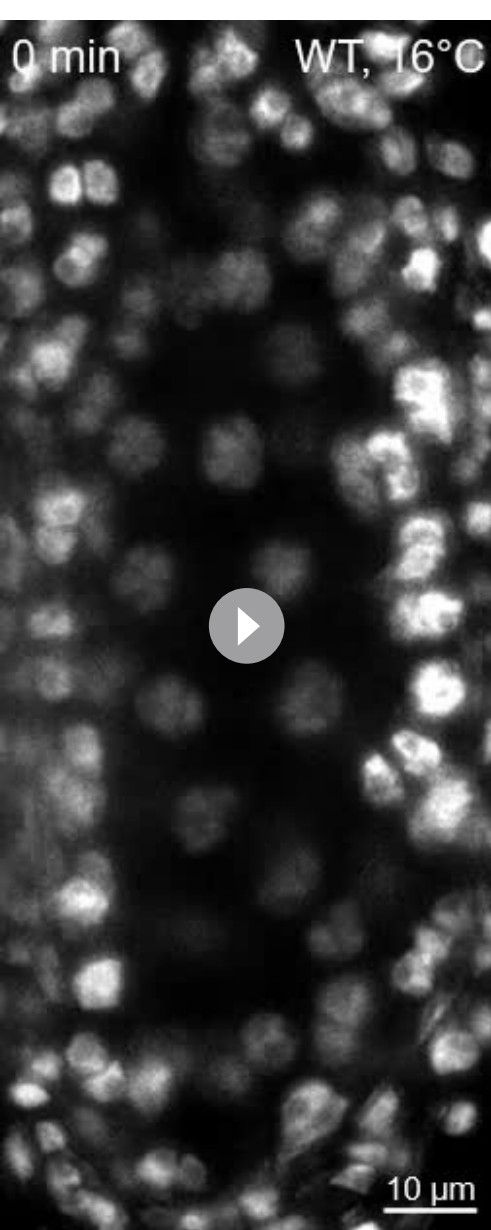

**Video 1.** Live imaging of meiosis in plants carrying HTA10:RFP chromatin marker grown at 16°C.
https://elifesciences.org/articles/90253/figures#video1

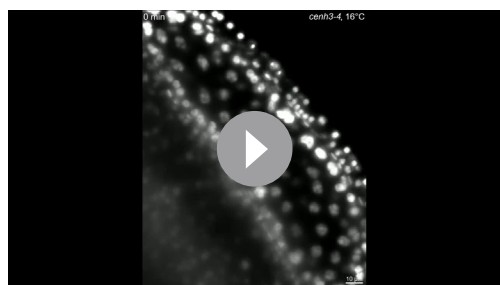

**Video 2.** Live imaging of meiosis in *cenh3-4* plants carrying HTA10:RFP chromatin marker grown at 16°C. Production of micronuclei could be seen in a few pollen mother cells.
https://elifesciences.org/articles/90253/figures#video2

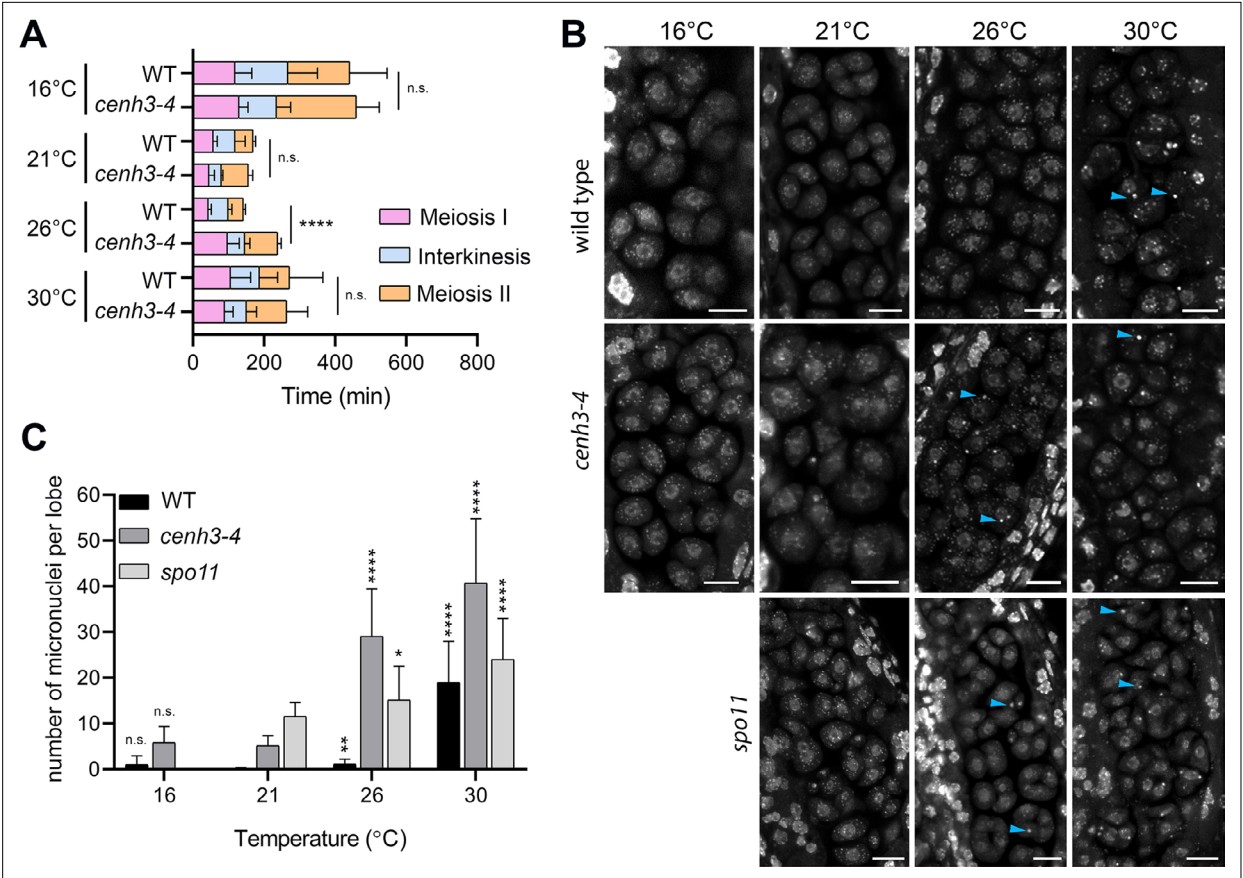

**Figure 2.** Effect of temperature on meiosis duration and micronuclei formation. (**A**) Graphical representation of the duration of meiosis I (from the end of diakinesis to the end of anaphase I; *Figure 2—figure supplement 1*), interkinesis, and meiosis II (prometaphase II to telophase II) calculated from live imaging of anthers in wild type (WT) and *cenh3-4* plants grown at 16°C, 21°C, 26°C, and 30°C. Error bars represent standard deviation (from 16°C to 30°C: in wild type n=36, 36, 36, 45 and *cenh3-4* n=35, 24, 24, 4, resp.) Significance of the difference is indicated (two-tailed t-test; ****p<0.0001). (**B**) Anther loculi in the tetrad stage of wild type, *cenh3-4* and *spo11-2-3* plants grown at 16°C, 21°C, 26°C, and 30°C. DNA was stained with DAPI. Blue arrowheads indicate examples of produced micronuclei. Scale bar = 10 μm. (**C**) Number of micronuclei per lobe in wild type (WT, n=19, 19, 19, 19), *cenh3-4* (n=19, 19, 19, 19), and *spo11-2-3* (n=19, 26, 21) plants. Error bars represent standard deviation. Significance of the difference from plants of the corresponding genotype grown at 21°C is indicated (two-tailed t-test; *p<0.05, **p<0.01, ***p<0.001, ****p<0.0001). Source values for (**A**) and (**C**) are available in *Source data 1*.

The online version of this article includes the following figure supplement(s) for figure 2:

**Figure supplement 1.** An example of a time lapse series of HTA10:RFP meiocytes indicating the meiotic stages used for calculating the duration of meiosis.

**Figure supplement 2.** Effect of changing night-day temperatures on wild type and *cenh3-4* plants.

et al., 2006) and its signal is most visible in pachytene when homologous chromosomes are fully synapsed.

To assess the effect of temperature on centromeres, we analyzed the CENH3 signal on pachytene chromosomes in anthers of the *Arabidopsis eYFP:CENH3* reporter line (*Le Goff et al., 2020*; *Demidov et al., 2022*) grown at 21°C, 26°C, and 30°C. While eYFP:CENH3 was readily detectable on pachytene chromosomes at 21°C, the signal decreased significantly at 26°C and declined further at 30°C (*Figure 3A and B*). This trend was observed in three independent experiments (*Figure 3B*). In contrast, the eYFP:CENH3 signal in tapetum nuclei adjacent to pollen mother cells (PMCs) exhibited the opposite trend and increased with elevated temperature (*Figure 3A and C*). This resulted in a striking difference in signal intensity at 30°C, where the tapetum exhibited a strong eYFP:CENH3 signal, whereas the signal on pachytene chromosomes in the same field of view was barely detectable (*Figure 3A*). We validated the observation that elevated temperature reduces the

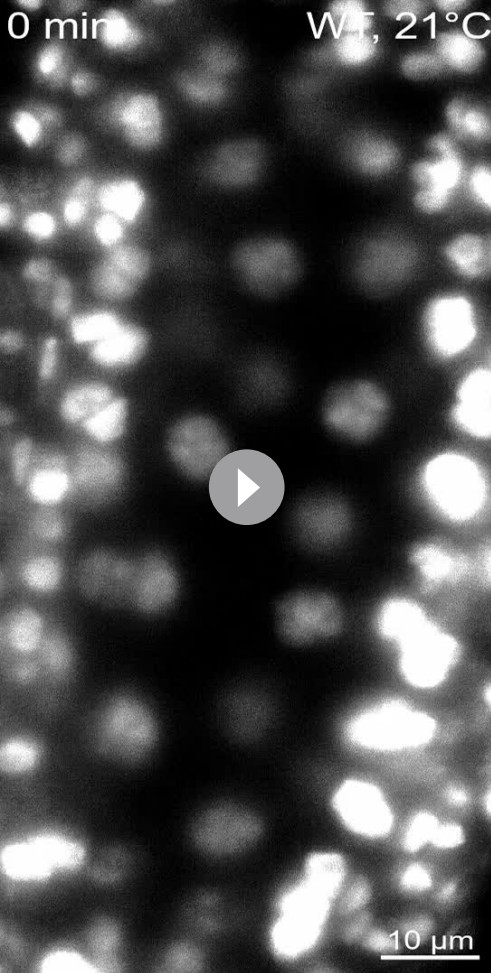

**Video 3.** Live imaging of meiosis in plants carrying HTA10:RFP chromatin marker grown at 21°C.
https://elifesciences.org/articles/90253/figures#video3

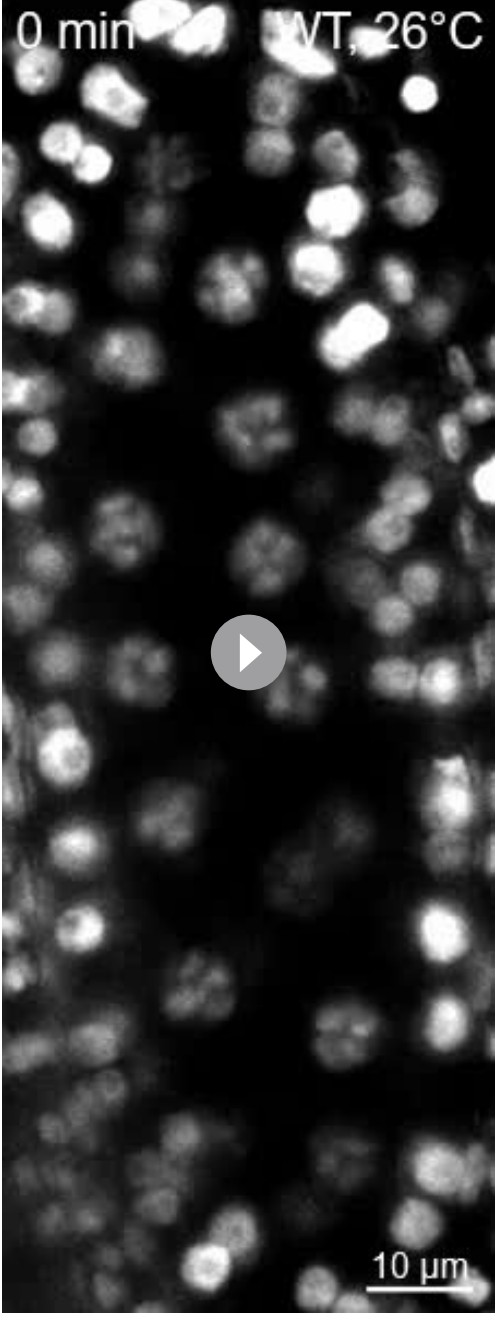

**Video 4.** Live imaging of meiosis in plants carrying HTA10:RFP chromatin marker grown at 26°C.
https://elifesciences.org/articles/90253/figures#video4

level of CENH3 on pachytene chromosomes also in non-tagged wild type plants by immunodetection using a CENH3 antibody (*Figure 3—figure supplement 1*).

CENH3 acts as a platform for the binding of additional kinetochore and SAC proteins. BMF1 is an *Arabidopsis* BUB1-related protein that associates with centromeres throughout the cell cycle (*Komaki and Schnittger, 2017*). We generated *Arabidopsis* marker lines expressing *BMF1::BMF1:eYFP* and *BMF1::BMF1:TagRFP* constructs, and validated BMF1 co-localization with CENH3 from early prophase I to telophase II (*Figure 3—figure supplement 2*, *Video 9*). The BMF1 signal was not detected in roots or in meiotic cells of *cenh3-4* mutants indicating that BMF1 loading on centromeres is CENH3 dependent (*Figure 3—figure supplement 3*). Analysis of the BMF1:eYFP signal in wild type plants grown at elevated temperatures showed a decrease in the signal on pachytene chromosomes at 26°C, and BMF1 was undetectable at 30°C (*Figure 3D and E*). These data suggest that similar to CENH3, elevated temperature reduces the kinetochore protein BMF1 at meiotic centromeres.

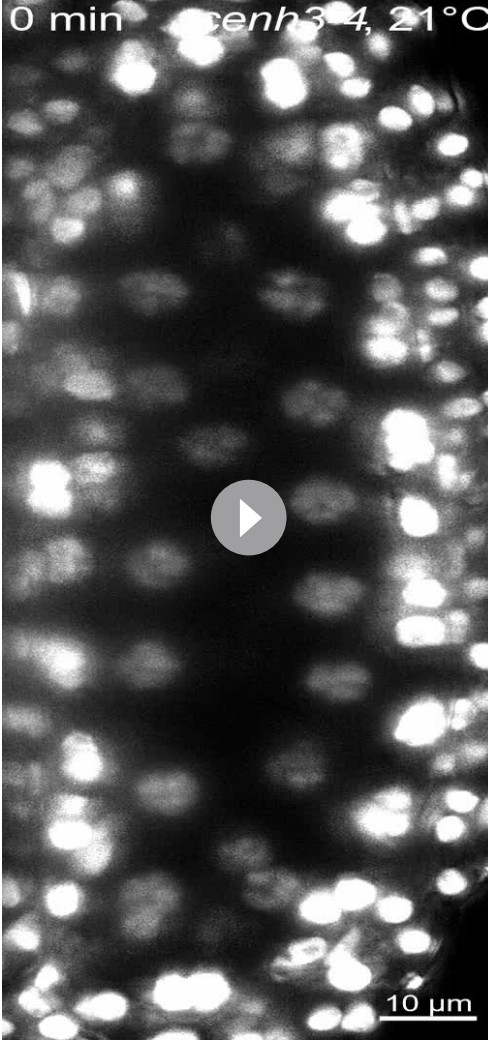

**Video 5.** Live imaging of meiosis in *cenh3-4* plants carrying HTA10:RFP chromatin marker grown at 21°C. Production of micronuclei could be detected in some pollen mother cells.

https://elifesciences.org/articles/90253/figures#video5

## Heat stress prolongs SAC in metaphase I

In our previous report, we showed that reduced amount of CENH3 and smaller centromeres prolong the biorientation of mitotic chromosomes in *cenh3-4* plants (*Capitao et al., 2021*). Biorientation is monitored by the SAC and its satisfaction triggers anaphase. We hypothesized that meiotic chromosomes with heat-induced reductions in CENH3 might take longer to properly attach to the spindle and satisfy the SAC. The core SAC proteins temporarily associate with the kinetochore during spindle formation and disappear just before the onset of anaphase.

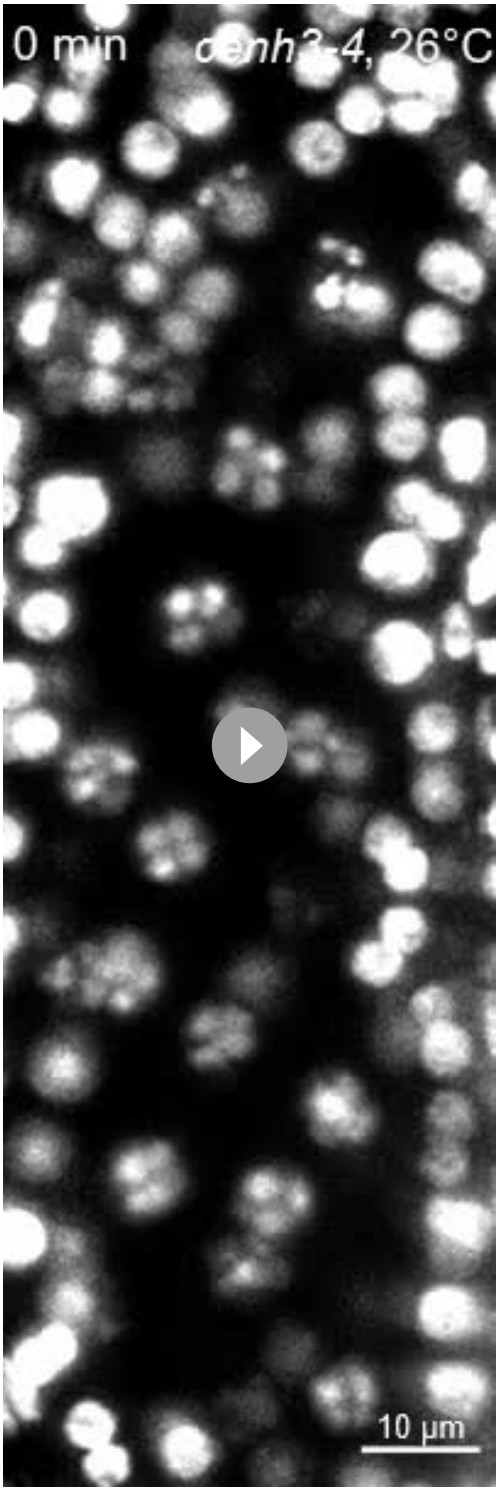

**Video 6.** Live imaging of meiosis in *cenh3-4* plants carrying HTA10:RFP chromatin marker grown at 26°C. Production of micronuclei could be detected in most pollen mother cells.

https://elifesciences.org/articles/90253/figures#video6

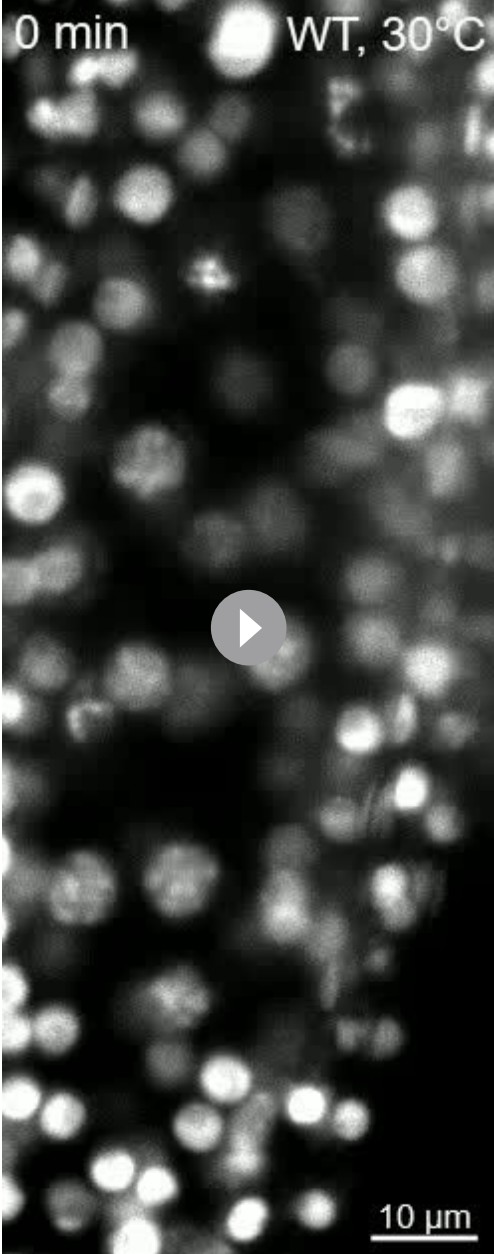

**Video 7.** Live imaging of meiosis in plants carrying HTA10:RFP chromatin marker grown at 30°C. Aberrant meiotic products are detected.

https://elifesciences.org/articles/90253/figures#video7

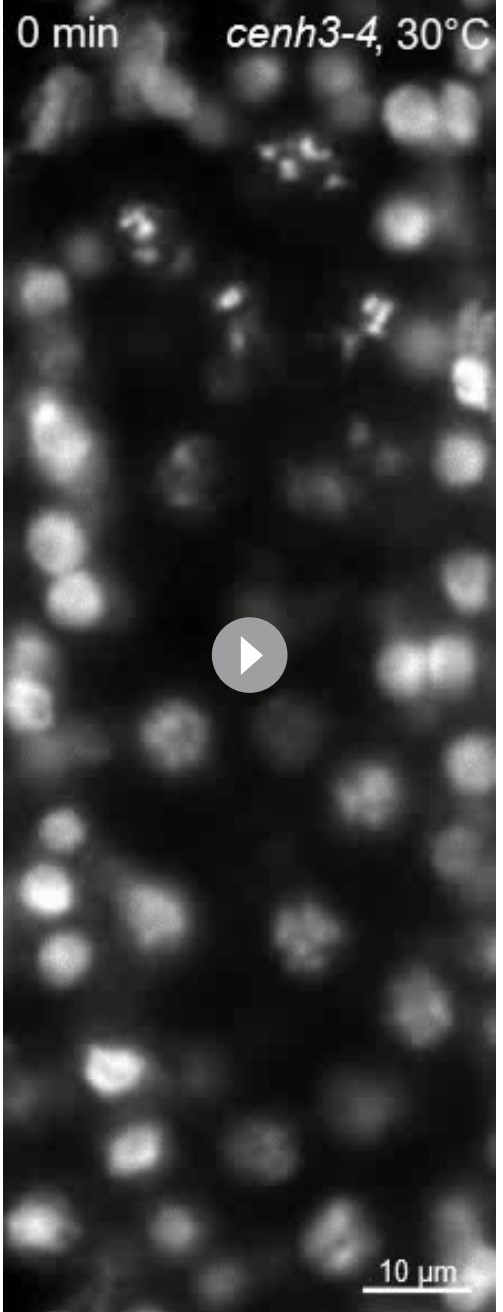

**Video 8.** Live imaging of meiosis in *cenh3-4* plants carrying HTA10:RFP chromatin marker grown at 30°C. Most pollen mother cells (PMCs) undergo aberrant meiotic division and only a few PMCs undergo normal meiotic division resulting in forming unbalanced tetrads.

https://elifesciences.org/articles/90253/figures#video8

BMF3 is one of the core components of the *Arabidopsis* SAC, which associates with the kinetochores during prometaphase and extends this association in the presence of a spindle inhibitor (*Komaki and Schnittger, 2017*; *Lampou et al., 2023*). To monitor the SAC on meiotic chromosomes, we generated *Arabidopsis* lines expressing *BMF3::BMF3:TagRFP,* and *BMF3::BMF3:GFP* together with the tubulin marker TagRFP:TUB4. First, we validated the localization of BMF3 on meiotic chromosomes by live cell imaging, as BMF3 has previously only been analyzed in the context of mitosis (*Komaki and Schnittger, 2017*). We observed the BMF3:GFP signal during prometaphase/metaphase I, it disappeared at the onset of anaphase I

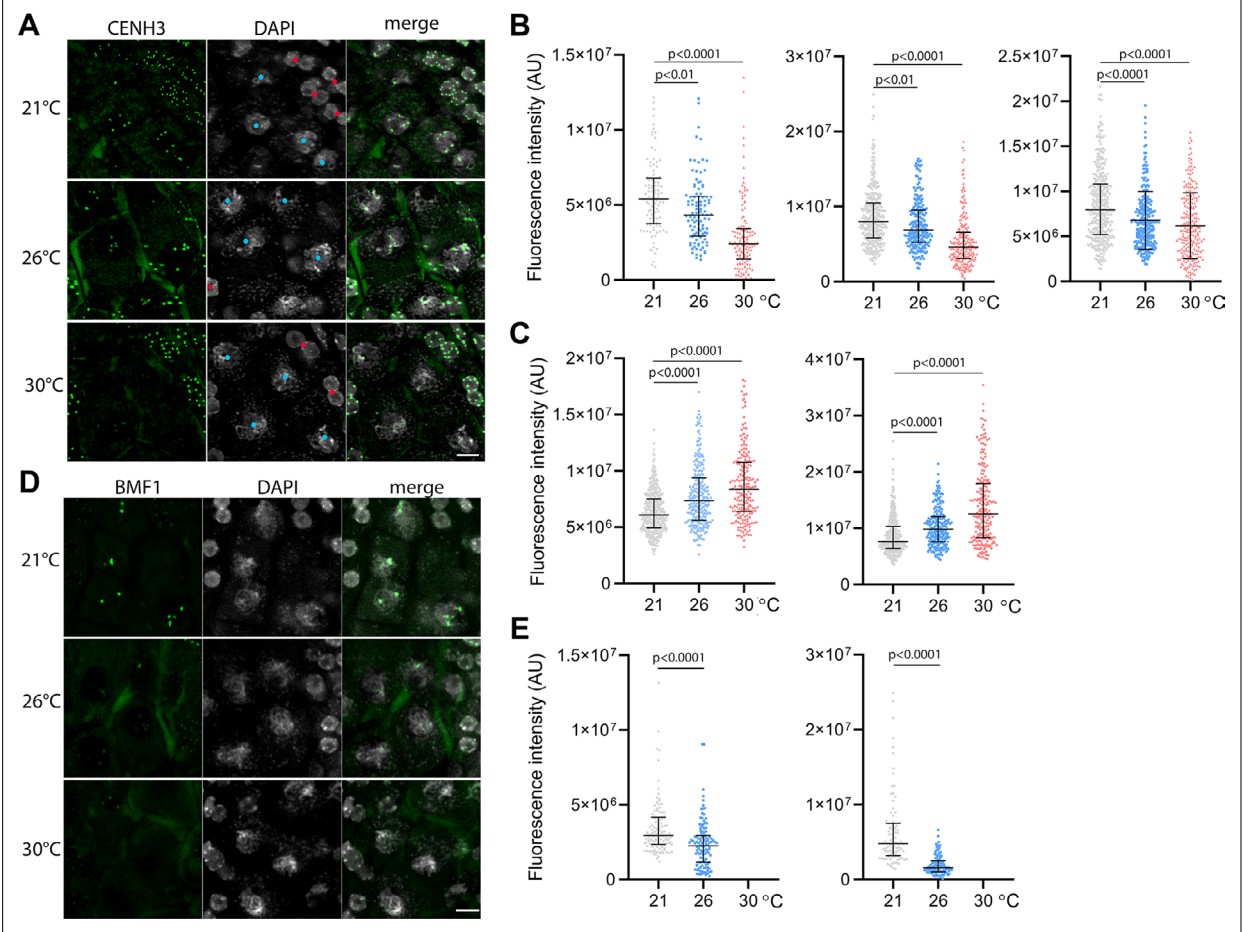

**Figure 3.** Effect of high temperature on centromere structure in wild type plants. (**A**) eYFP:CENH3 expression and DAPI staining of meiotic pachytene (blue dots) and mitotic tapetal cells (red dots) in wild type plants grown at 21°C, 26°C, and 30°C. Scale bar = 5 μm. (**B**) Quantification of CENH3 fluorescence intensity per centromere in pachytene. Each interleaved scatter plot with median and interquartile range shows results of an independent experiment (left graph n=102, 103, 125, middle n=359, 220, 233 and right graph n=386, 268, 233). (**C**) Quantification of eYFP:CENH3 signal intensity in tapetum cells of plants grown at 21°C, 26°C, and 30°C. Each graph represents an independent experiment (left graph n=369, 262, 203 and right graph n=358, 266, 213). (**D**) Expression of BMF1:eYFP in pachytene in plants grown at 21°C, 26°C, and 30°C. DNA is counterstained with DAPI. Scale bar = 5 μm. (**E**) Quantification of BMF1:eYFP signal intensity per centromere in pachytene in two independent experiments; left graph n=120 and right n=100. Two-tailed t-test is used to depict the significance of the difference. Source values for (**B**, **C**, and **E**) are available in *Source data 1*.

The online version of this article includes the following figure supplement(s) for figure 3:

**Figure supplement 1.** Immunodetection of CENH3 on meiotic chromosomes.

**Figure supplement 2.** Co-localization of the kinetochore BMF1:TagRFP (red) and centromeric eYFP:CENH3 (yellow) proteins through meiosis in DAPI stained (gray) meiocytes.

**Figure supplement 3.** Association of BMF1 with centromeres in *cenh3-4* plants.

---

and reappeared again during metaphase II, suggesting that BMF3 is a component of the meiotic SAC (*Figure 4A*, *Video 10*). We then analyzed the effect of temperature on SAC duration. Since the signal was more prominent in metaphase I, we measured the duration of the active SAC during meiosis I. We observed that while the BMF3:GFP signal persisted for an average approximately 22.7 min at 21°C and 26°C, its appearance was prolonged to 40.5 min at 30°C (*Figure 4B*, *Videos 11 and 12*). We also noticed that the intensity of the BMF3 signal appeared to decrease with increasing temperature (*Figure 4—figure supplement 1A*). A weaker but prolonged association of BMF3 with the meiosis I kinetochore was also observed in *cenh3-4* mutants with reduced levels of centromeric histone (*Figure 4—figure supplement 1B and C*). These observations are consistent with the notion that elevated temperature leads to partial depletion of CENH3 and impairment of centromere structure, which may prolong the time required for chromosome biorientation and SAC satisfaction.

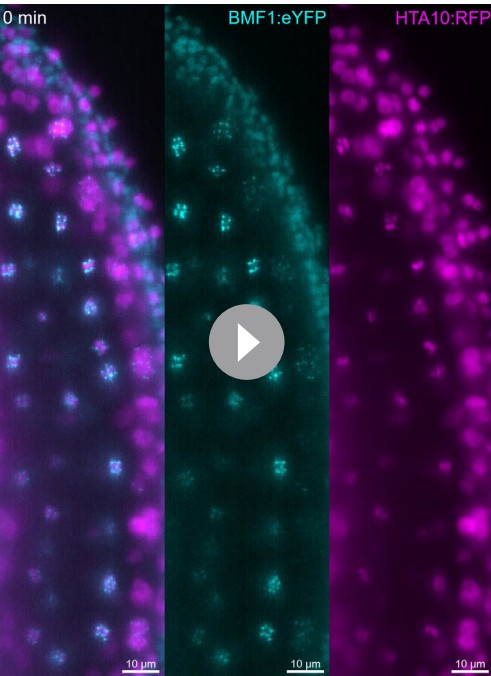

**Video 9.** Live imaging of BMF1:eYFP kinetochore marker (cyan) and HTA10:RFP chromatin marker (magenta) during meiosis.

https://elifesciences.org/articles/90253/figures#video9

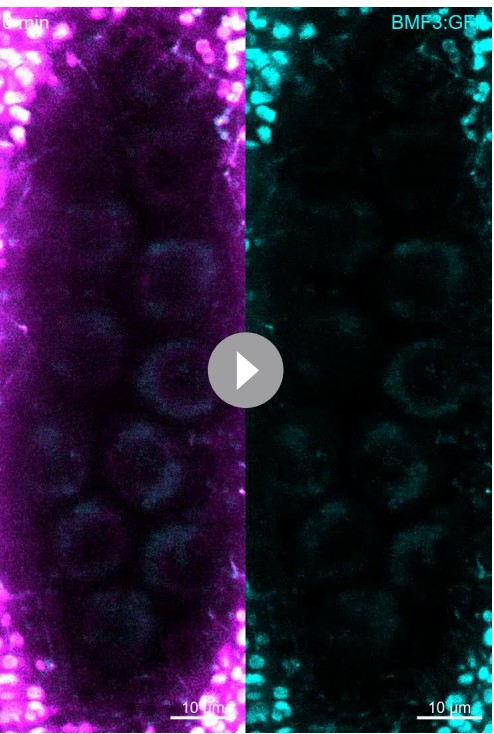

**Video 10.** Live imaging of BMF3:GFP spindle assembly checkpoint marker (cyan) and TagRFP:TUB4 tubulin marker (magenta) during both meiosis.

https://elifesciences.org/articles/90253/figures#video10

## Discussion

In this study we investigated the impact of increased temperature on fertility and meiotic progression in *Arabidopsis*. We observed that *A. thaliana* (ecotype Col-0) exhibited the highest fertility when grown at 16°C, which closely resembles the average temperature during flowering in its natural habitats (*Brachi et al., 2010*). At this

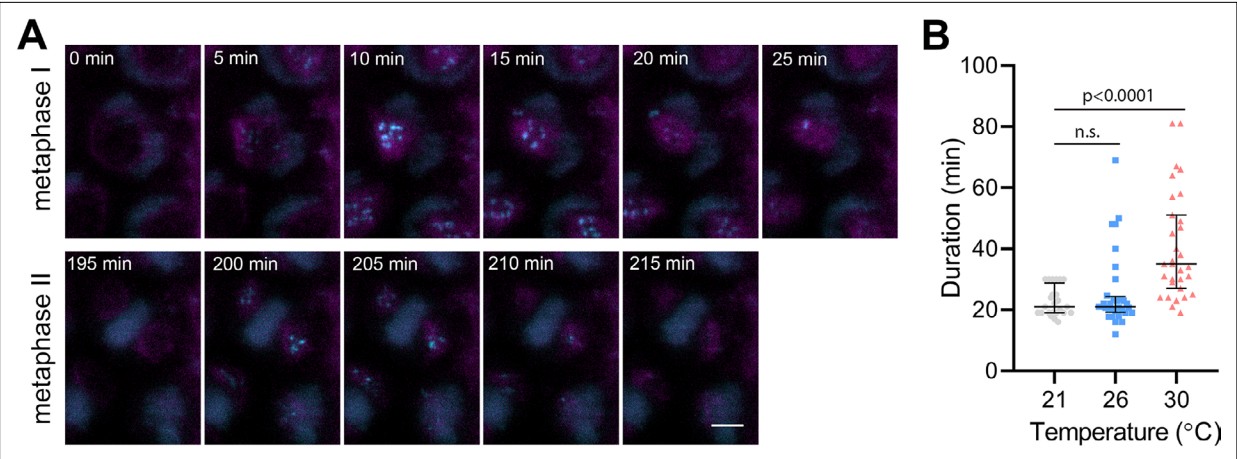

**Figure 4.** Effect of high temperature on the duration of BMF3:GFP localization during wild type meiosis. (**A**) Time lapse series of BMF3:GFP (cyan) and TagRFP:TUB4 (magenta) in pollen mother cell from nuclear envelope breakdown to telophase II. Scale bar = 5 µm. (**B**) Duration of BMF3:GFP signal in plants grown at 21°C (n=24), 26°C (n=32), and 30°C (n=31). Significance of the difference was calculated via two-tailed t-test. Source values for (**B**) are available in *Source data 1*.

The online version of this article includes the following figure supplement(s) for figure 4:

**Figure supplement 1.** Association of BMF3 with centromeres in *cenh3-4* plants.

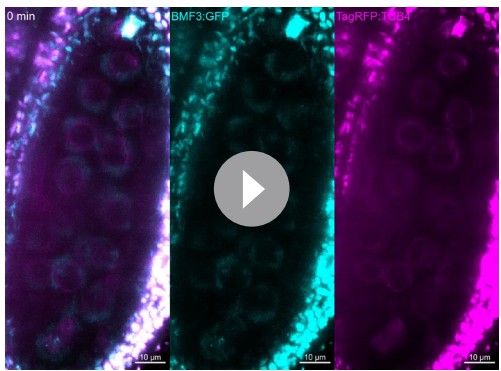

**Video 11.** Live imaging of BMF3:GFP spindle assembly checkpoint marker (cyan) and TagRFP:TUB4 tubulin marker (magenta) during first meiotic division of plants grown at 26°C.

https://elifesciences.org/articles/90253/figures#video11

temperature, meiotic divisions took more than twice as long as at 21°C. *Arabidopsis* grown at 17°C was reported to exhibit slower microsporogenesis, with the entire meiosis process lasting 48 hr as opposed to 32 hr at 24°C (*Zhu et al., 2020*). Interestingly, the slower progression was found to be less detrimental for several mutations that affect microsporogenesis, indicating that a slower pace provides greater robustness. Overall, these observations suggest that the pace of meiosis in *Arabidopsis* is well optimized for its natural growth conditions, enabling the plants to achieve maximum fitness.

The process of meiosis is progressively accelerated with increasing temperature (*De Jaeger-Braet et al., 2022*; *Figure 2*). However, the speed of meiotic divisions reaches its maximum at 26°C and slows down at 30°C under the conditions used in this study. At 30°C, fertility is lost and high incidence of micronuclei is observed, implying that the temperature around 30°C impairs some molecular processes involved in chromosome segregation and genome integrity. The micronuclei can represent acentric fragments derived from aberrant processing of recombination intermediates. Indeed, the impact of temperature on recombination machinery is well documented. A moderate increase in temperature leads to shift in chiasma distribution and cross-over frequency (*Higgins et al., 2012*; *Lloyd et al., 2018*; *Modliszewski et al., 2018*). Heat stress above 30°C severely impairs recombination and synapsis, leading to univalents and chromosome segregation defects (*De Storme and Geelen, 2020*; *Ning et al., 2021*; *De Jaeger-Braet et al., 2022*; *Fu et al., 2022*). Nevertheless, we observed an increased incidence of micronuclei also in SPO11-deficient plants that do not form meiotic DNA double-strand breaks indicating that only a portion of micronuclei is derived from aberrant recombination.

Micronuclei can also be formed by aberrant chromosome segregation during anaphase (*Fenech et al., 2011*). The accurate partitioning of chromosomes depends on the correct attachment of centromeres to the spindle microtubules. Recent reports have indicated that the stability of microtubules in plant meiocytes is compromised by heat stress (*Wang et al., 2017*; *De Jaeger-Braet et al., 2022*). In this study, we have demonstrated that high temperature also affects the structure and function of meiotic centromeres. Our findings suggest that elevated temperature reduces the amount of CENH3 and BMF1 on meiotic centromeres (*Figure 3*). These weakened centromeres may be less effective in establishing productive interactions with spindle microtubules. This is consistent with the prolonged SAC we observed through the longer residency of BMF3 on prometaphase I centromeres at 30°C (*Figure 4*). Taken together, our data indicate that heat stress impairs centromere function, which in turn decreases the efficiency of proper attachment of chromosomes to the meiotic spindle. As a result, these chromosomes may not be transported to their intended destination in a timely manner and, therefore, not incorporated into the newly formed nuclei. This scenario also explains the increased temperature sensitivity of *cenh3-4* plants, which already have smaller centromeres (*Capitao et al., 2021*); even moderately elevated temperature may further enhance this defect.

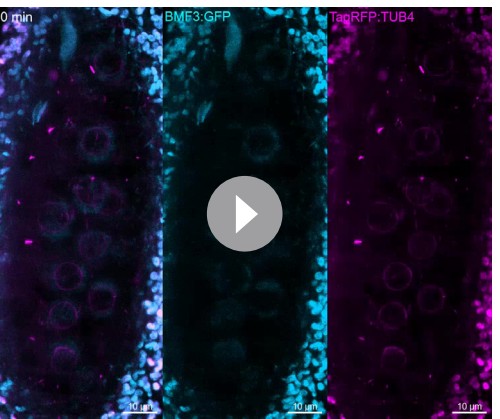

**Video 12.** Live imaging of BMF3:GFP spindle assembly checkpoint marker (cyan) and TagRFP:TUB4 tubulin marker (magenta) during the first meiotic division of plants grown at 30°C.

https://elifesciences.org/articles/90253/figures#video12

How elevated temperature affects the amount of CENH3 on centromeres? Studies in human cells have revealed that CENH3 nucleosomes are less stable than previously thought. The amount of CENH3 at centromeres declines over time and must be actively replenished to preserve centromere identity and proliferative potential (*Swartz et al., 2019*). Transcription has been shown to cause the eviction of pre-existing CENH3 from the centromeres (*Swartz et al., 2019*). In *Arabidopsis*, depletion of CENH3 from centromeres has been observed in leaf cells with increasing age, indicating that this phenomenon also occurs in plants (*Lermontova et al., 2011b*). The *Arabidopsis* centromeric satellite repeat CEN180 undergoes pervasive transcription, which is largely repressed by epigenetic silencing mechanisms (*May et al., 2005*). However, heat stress can alleviate the repression and induce transcription of CEN180 from silent loci (*Tittel-Elmer et al., 2010*). This temperature-induced transcription can increase the eviction of CENH3 nucleosomes, potentially compromising centromere structure if not replenished. Interestingly, in contrast to PMCs, tapetum cells do not exhibit CENH3 loss at elevated temperatures (*Figure 3C*). This may be due to the different efficiency of CENH3 deposition to replenish the heat-induced loss of CENH3 in these cell types. Accordingly, *Arabidopsis* meiotic cells possess a specialized CENH3 loading mechanism that seems more stringent than the loading mechanism operating in mitotic cells (*Lermontova et al., 2011a*; *Ravi et al., 2011*; *Schubert et al., 2014*).

Crosses between wild type and plants with altered CENH3 can result in postzygotic loss of one set of parental chromosomes (*Ravi and Chan, 2010*). This is attributed to inefficient re-loading of CENH3 on the set of parental chromosomes with the altered centromeres in hybrid embryos (*Marimuthu et al., 2021*). Recent studies showed that heat stress applied during early embryogenesis could enhance the centromere-mediated genome elimination in *Arabidopsis* (*Ahmadli et al., 2023*; *Jin et al., 2023*; *Wang et al., 2023*). This indicates that similarly to meiocytes, elevated temperature can also destabilize centromeres in embryonic cells.

In conclusion, our study highlights the significant impact of temperature on the centromere structure and reproduction in *Arabidopsis*. We found that already moderate increase in temperature has a discernable effect on pollen production and silique length, and exposure to 30°C impairs the structure of centromeres and leads to plant sterility. These findings are particularly relevant in the context of climate change, where rising temperatures and more frequent weather extreme periods pose a threat to global food security by disrupting plant reproductive processes. As such, our study provides important insights into the mechanisms that contribute to the reduction of plant fertility in response to elevated temperature. Further research on molecular aspects underlying these effects may help to develop strategies to generate plants more resilient to extreme weather during their reproductive phase.

## Materials and methods

### Plant material and growth conditions

*A. thaliana* ecotype Columbia (Col-0), *cenh3-4* (*Capitao et al., 2021*), and *spo11-2-3* (*Hartung et al., 2007*) seeds were grown on soil in growth chambers at 21°C, 16 hr/8 hr light/dark cycles and 50% of humidity until the 1.04 growth stage. Plants were then transferred to different chambers with continuous 16°C, 21°C, 26°C, 30°C or 34°C/18°C and 21°C/18°C at 16 hr/8 hr light/dark cycles and 50% of humidity. Plants used for live cell imaging were generated by crossing *HTA10:RFP* (*Valuchova et al., 2020*) and *pRPS5A::TagRFP:TUB4* (*Prusicki et al., 2019*) reporter lines with *cenh3-4* mutant and *BMF3::BMF3:GFP* reporter lines, respectively. *eYFP:CENH3* reporter line was described and characterized in previous studies (*Le Goff et al., 2020*; *Demidov et al., 2022*).

### Generating reporter lines

To generate the BMF1 and BMF3 reporter lines, we amplified the promoter and genomic regions of BMF1 using the primers CACCTGAGTCTCCAACGTTA and CGAAGAGCATAACGAGATGCG. The PCR products were then cloned into the destination vectors pGWB659 and pGWB640, respectively, to create marker lines tagged with TagRFP and eYFP at the C-terminus. Similarly, we amplified the BMF3 promoter and genomic regions using the primers CACCATGCAGATGGTCCTCC and GAAGTCCATTGGCATTGCAAA, and subcloned them into pGWB459 and pGWB650 destination vectors using gateway cloning to generate the *BMF3::BMF3:TagRFP* and *BMF3::BMF3:GFP* lines, respectively. We then introduced these constructs into wild type and *cenh3-4* plants using *Agrobacterium*-mediated

floral dip transformation. We used non-segregating homozygous transgenic lines for microscopic analyses.

## Assessment of plant fertility

Pollen viability was determined by Alexander staining (*Alexander, 1969*). Anthers were imaged using Zeiss Axioscope A1 equipped with 20×/0.5 objective (Zeiss) and processed in Zeiss ZEN software. Silique length was assessed from images of main stems scanned by an Epson scanner and the siliques were measured using the Fiji Analyze/Measure function (*Schindelin et al., 2012*).

## Cytology

To examine micronuclei, DAPI staining of PMCs was used in whole anther as described (*Capitao et al., 2021*). Anthers were imaged using Zeiss LSM780 confocal microscope (63×/1.4 oil objective) and image processing was done using Zeiss ZEN software (Zeiss). To quantify the fluorescence signal intensity of eYFP:CENH3 and BMF1:eYFP, anthers were DAPI stained as described above, and Z-stacks were acquired using Zeiss LSM880 confocal microscope equipped with the Fast module 32-channels Airyscan detector (63×/1.4 oil objective). The fluorescence intensity was quantified using Fiji (*Schindelin et al., 2012*) according to the procedure described by *Shihan et al., 2021*. This involved generating a SUM of signal in Z-stacks covering one nucleus, background subtraction, and measuring Raw Integral Density per one signal. This process was repeated individually in each nucleus.

Immunodetection of CENH3 was performed with a custom-made polyclonal antiserum raised against the N-terminal peptide of CENH3 (1:1000) (*Capitao et al., 2021*) and anti-Rabbit-CY3 (Jackson ImmunoResearch). *Arabidopsis* inflorescences were fixed in 1× PBS buffer supplemented with 4% formaldehyde and 0.05% Tween for 15 min in vacuum and 45 min at room temperature. The floral buds were washed with 1× PBS and digested with cytohelicase (0.1 g cytohelicase, 0.25 g polyvinylpyrrollidone, and 0.375 g sucrose in 25 ml of water) for 2 hr. Buds were washed once with 1× PBS and anthers were dissected on glass slides, squashed and frozen in liquid nitrogen. The slides were blocked with 3% BSA in 1× PBS supplemented with 0.5% Triton X-100 for 30 min at 37°C. Anti-CENH3 antibody in 1× PBS supplemented with 3% BSA was added to each slide and incubated over night at 4°C. Slides were washed with 1× PBS and anti-Rabbit-CY3 secondary antibody diluted in 1× PBS supplemented with 3% BSA was added to each slide and incubated for 1 hr at 37°C. Slides were washed with 1× PBS, stained with DAPI, and mounted in Vectashield. Slides were observed using AxioImager.Z2 fluorescence microscope (Zeiss). Images were analyzed using ZEN (Zeiss) and Fiji (*Schindelin et al., 2012*) softwares. Signal intensity was measured at the peak signal of each centromere and was normalized to the background signal.

## Live cell imaging

Live cell imaging of meiosis was performed by light-sheet fluorescence microscopy using Light-sheet Z.1 microscope (Zeiss) as previously described (*Valuchova et al., 2020*; *Capitao et al., 2021*). Imaging of male meiosis was conducted using 10× or 20× objectives (Detection Optics 10×/0.5 or 20×/1.0), single illumination (Illumination Optics 10×/0.2) and two track imaging with 488 nm laser for GFP and eYFP, and 561 nm laser for RFP, in 5 min time increments. Imaging of BMF3:GFP and TagRFP:TUB4 markers for SAC analysis was performed using fast scanning in 1 min time increments. Images were deconvolved with a Regularized Inverse Filter and further processed in Zeiss ZEN software for Light-sheet (Zeiss). To correct occasional sample drift, the Correct 3D drift plugin in Fiji (*Parslow et al., 2014*) was used.

## Acknowledgements

We acknowledge the support from CEITEC MU Core facilities Plant Sciences and CELLIM, supported by the Czech-Bioimaging (No. LM2018129) infrastructure project funded by MEYS CZ. This work was funded by the Czech Science Foundation (grant 21-25163J to KR). MK is supported by Deutsche Forschungsgemeinschaft (LE2299/5-1).

# Additional information

## Funding

| Funder | Grant reference number | Author |
|---|---|---|
| Czech Science Foundation | 21-25163J | Karel Riha |
| Deutsche Forschungsgemeinschaft | LE2299/5-1 | Inna Lermontova |
| Ministry of Education, Youth and Sports, Czech Republic | LM2018129 | Karel Riha |

The funders had no role in study design, data collection and interpretation, or the decision to submit the work for publication.

## Author contributions

Lucie Crhak Khaitova, Conceptualization, Formal analysis, Investigation, Visualization, Methodology, Writing - original draft, Writing - review and editing; Pavlina Mikulkova, Formal analysis, Investigation, Visualization, Methodology; Jana Pecinkova, Resources, Investigation, Methodology; Manikandan Kalidass, Investigation, Methodology; Stefan Heckmann, Resources; Inna Lermontova, Conceptualization, Resources, Supervision, Funding acquisition; Karel Riha, Conceptualization, Formal analysis, Supervision, Funding acquisition, Writing - original draft, Writing - review and editing

## Author ORCIDs

Lucie Crhak Khaitova ![ORCID] http://orcid.org/0000-0003-1195-593X
Jana Pecinkova ![ORCID] http://orcid.org/0000-0003-0006-5448
Stefan Heckmann ![ORCID] http://orcid.org/0000-0002-0189-8428
Karel Riha ![ORCID] http://orcid.org/0000-0002-6124-0118

Reviewer #2 (Public Review): https://doi.org/10.7554/eLife.90253.3.sa1
Reviewer #3 (Public Review): https://doi.org/10.7554/eLife.90253.3.sa2
Author response https://doi.org/10.7554/eLife.90253.3.sa3

# Additional files

## Supplementary files

- MDAR checklist

- Source data 1. Raw source data of measurements used for making graphs and statistics.

## Data availability

All data generated or analysed during this study are included in the manuscript and supporting files.

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
