## [Editor Report · eLife assessment]

This study is an **important** contribution to our insights into the impact of heat stress on sexual reproduction in plants and provides information about how centromere integrity is affected by heat stress during male meiosis in *Arabidopsis thaliana*. The evidence supporting the claims, specifically the dynamics of tagged proteins in meiocytes by live cell imaging is **solid**, even though a deeper mechanistic understanding is still lacking.

---

## [Referee Report · Reviewer #2 (Public Review)]

Summary:

Here the authors examine how increased temperature affects pollen production and fertility of *Arabidopsis thaliana* plants grown at selected temperature conditions ranging from 16C to 30C. They show that pollen production and fertility decline with increasing temperature. To identify the cause of reduced pollen and fertility, they resort to living cell imaging of male meiotic cells to identify that duration of meiosis increases with an increase in temperature. They also show that pollen sterility is associated with the increased presence of micronuclei likely originating from heat stress-induced impaired meiotic chromosome segregation. They correlate abnormal meiosis to weakened centromere caused by meiosis-specific defective loading of the centromere-specific histone H3 variant (CenH3) to the meiotic centromeres. Similar is the case with kinetochore-associated spindle assembly checkpoint(SAC) protein BMF1. Intriguingly, they observe a reverse trend of strong CENH3 presence in the somatic cells of the tapetum in contrast to reduced loading of CENH3 in male meiocytes with increasing temperature. In contrast to CENH3 and BMF1, the SAC protein BMF3 persists for longer periods than the WT control, based on which authors conclude that the heat stress prolongs the duration of SAC at metaphase I, which in turn extends the time of chromosome biorientation during meiosis I. This study provides insights onto the processes that affect plant reproduction with increasing temperatures which may be relevant to develop climate-resilient cultivars.

Strengths:

This study shows that the centromere function is affected under heat stress in meiotic cells by modulating the dynamics of the centromere specific histone H3 (CENH3) that in turn compromises the assembly of kinetochore complex proteins. This they have demonstrated by the way of live cell imaging of male meiocytes by tracking the loading dynamics and resident time of fluorescently tagged centromere/kinetochore proteins and spindle assembly checkpoint proteins.

Weaknesses:

Though the results presented here are interesting and solid, the current study lacks a deeper mechanistic understanding of what causes the defective loading of CenH3 to the centromeres, and why the SAC protein BMF3 persists only at meiotic centromeres to prolong the spindle assembly checkpoint, which will be interesting to delve further to completely understand the process.

Here the authors monitor one representative centromere protein CENH3, one kinetochore-associated SAC protein BMF1, and the SAC protein BMF3 to conclude that heat stress impairs centromere/kinetochore function and prolongs SAC with increased temperatures. Centromere and its associated protein complex the kinetochores and the SAC contains a multitude of proteins, some of which are well characterized in *Arabidopsis thaliana*. Hence the authors could have used additional such tagged proteins to further strengthen their claim.

---

## [Referee Report · Reviewer #3 (Public Review)]

Summary:

Khaitova et al. report the formation of micronuclei during Arabidopsis meiosis under elevated temperature. Micronuclei form when chromosomes are not correctly collected to the cellular poles in dividing cells. This happens when whole chromosomes or fragments are not properly attached to the kinetochore microtubules. The incidence of micronuclei formation is shown to increase at elevated temperature in wild type and more so in the weak centromere histone mutant cenH3-4. The number micronuclei formation at high temperature in the recombination mutant spo11 is like that in wild type, indicating that the increased sensitivity of cenh3-4 is not related to the putative role of cenh3 in recombination. The abundance of CENH3-GFP at the centromere declines with higher temperature and correlates with a decline in spindle assembly checkpoint factor BMF1-GFP at the centromeres. The reduction in CENH3-GFP under heat is observed in meiocytes whereas CENH3-GFP abundance increases in the tapetum, suggesting there is a differential regulation of centromere loading in these two cell types. These observations are in line with previous reports on haploidization mutants and their hypersensitivity to heat stress.

Strength:

The paper shows that the kinetochore function during meiosis is sensitive to high temperature and this leads to inequivalent chromosome segregation during meiosis and reduced fertility.

Weakness:

The increased sensitivity to high temperature stress of the hypomorphic mutant cenh3-4 mutant not only reduces fertility but also growth, which is not accompanied with the formation of micronuclei as in meiosis. The impact on mitosis therefore seems to be different from that in meiosis.

---

## [Author Response]

The following is the authors’ response to the original reviews.

Response to reviewers:

We would like to thank all the reviewers and the editors for their thorough and helpful feedback on our work. Before addressing specific questions and points, we would like to make a general comment on a mechanistic aspect of this study. The reviewers correctly pointed out that our study does not reveal the molecular mechanism that leads to centromeric histone depletion specifically from meiotic chromosomes. Identifying this mechanism requires a deep and thorough understanding of how centromeric histones are loaded and centromeres are established each cell cycle, and how they are maintained over time in different cell types. To our knowledge, these mechanisms have not been described in plants. To add a further layer of complexity, it appears that the mechanisms governing CENH3 maintenance may be (partially) different in plant mitotic and meiotic cells, and the mechanistic basis of this difference is unknown. Obviously, these are interesting but also complex questions and their resolution will require considerable resources and effort, which we believe is beyond the scope of this manuscript. Nevertheless, our finding that CENH3 maintenance and centromere function in meiotic cells are sensitive to heat stress is an unexpected discovery with profound implications for plant adaptation, which provides a strong incentive for further exploration of centromere maintenance mechanisms in plants.

Furthermore, we would like to apologize to reviewers for poor quality of pictures in the original submission. It was decreased by conversion to a pdf format during submission.

**eLife assessment**
This important study reports how heat stress affects centromere integrity by compromising the loading of the centromere protein CENH3 and by prolonging the spindle assembly checkpoint during male meiosis in *Arabidopsis thaliana*. The evidence supporting the claims by live cell imaging is convincing, although deeper mechanistic insight is lacking, making the study overall somewhat preliminary in nature. This work will be of interest to a broad audience of biologists working on how chromatin states are affected by stress conditions.
**Public Reviews:**

**Reviewer #1 (Public Review):**
Summary:Khaitova and co-workers present here an analysis of centromere composition and function during elevated temperatures in the plant Arabidopsis. The work relates to the ongoing climate change during which spikes in high temperatures will be found. Hence, the paper addresses a timely subject.The authors start by confirming earlier studies that high temperatures reduce the fertility of Arabidopsis plants. Interestingly, a hypomorphic mutant of the centromeric histone variant CENH3 (CENP-A), which was previously described by the authors, sensitizes plants to heat and results in a drop in viable pollen and silique length. The drop in fertility coincides with the formation of micronuclei in meiosis and an extension of meiotic progression as revealed by live cell imaging. Based on this finding, the authors then show that at high temperatures, the fluorescence intensity of a YFP:CENH3 declines in meiosis but remarkably not in the surrounding cells (tapetum cells). In addition, the amount of BMF1 (a Bub1 homolog and part of the spindle assembly checkpoint) also appears to decline on the kinetochores of meiocytes as judged by BMF1 reporter line. However, whether this is dependent on a decline of CENH3 or represents a separate pathway is not clear.

We provide new data in Figure S6 showing that BMF1 loading on centromeres is substantially reduced in cenh3-4 mutants. Thus, efficient tethering of BMF1 to centromeres depends on CENH3.

Finally, the authors measure the duration of the spindle checkpoint and find that it is extended under high temperatures from which they conclude that the attachment of spindle fibers to kinetochores is compromised under heat.Strengths:This is an interesting and important paper as it links centromere organization/function to heat stress in plants. A major conclusion of the authors is that weakened centromeres, presumably by heat, may be less effective in establishing productive interactions with spindle microtubules.Weaknesses:The paper does not explain the molecular reason why CENH3 levels in meiocyctes are reduced or why the attachment of spindle fibers to kinetochore is less efficient at high versus low temperatures.

While we cannot explain the molecular mechanism underlying temperature-dependent depletion of CENH3 in meiocytes, the less efficient attachment of microtubules to the kinetochores at higher temperatures is likely caused by reduced levels of CENH3, which result in smaller centromeres that are less effective in establishing productive microtubule-kinetochore attachments. Here (new Figure S6) and in our previous study (Capitao et al. 2021), we have shown that amount of centromere/kinetochore proteins is reduced at centromeres in cenh3-4 mutants, and that these plants exhibit prolonged SAC and slower chromosome biorientation.

**Reviewer #2 (Public Review):**
Summary:This work investigates how increased temperature affects pollen production and fertility of *Arabidopsis thaliana* plants grown at selected temperature conditions ranging from 16C to 30C. They report that pollen production and fertility decline with increasing temperature. To identify the cause of reduced pollen and fertility, they resort to living cell imaging of male meiotic cells to identify that the duration of meiosis increases with an increase in temperature. They also show that pollen sterility is associated with the increased presence of micronuclei likely originating from heat stress-induced impaired meiotic chromosome segregation. They correlate abnormal meiosis to weakened centromere caused by meiosis-specific defective loading of the centromere-specific histone H3 variant (CenH3) to the meiotic centromeres. Similar is the case with kinetochore-associated spindle assembly checkpoint(SAC) protein BMF1. Intriguingly, they observe a reverse trend of strong CENH3 presence in the somatic cells of the tapetum in contrast to reduced loading of CENH3 in male meiocytes with increasing temperature. In contrast to CENH3 and BMF1, the SAC protein BMF3 persists for longer periods than the WT control, based on which authors conclude that the heat stress prolongs the duration of SAC at metaphase I, which in turn extends the time of chromosome biorientation during meiosis I. The study provides preliminary insights into the processes that affect plant reproduction with increasing temperatures which may be relevant to develop climate-resilient cultivars.Strengths:The authors have mastered the live cell imaging of male meiocytes which is a technically demanding exercise, which they have successfully employed to examine the time course of meiosis in *Arabidopsis thaliana* plants exposed to different temperature conditions. In continuation, they also monitor the loading dynamics and resident time of fluorescently tagged centromere/kinetochore proteins and spindle assembly checkpoint proteins to precisely measure the time duration of respective proteins to study their precise dynamics and function in male meiosis.Weaknesses:Here the authors use only one representative centromere protein CENH3, one kinetochore-associated SAC protein BMF1, and the SAC protein BMF3 to conclude that heat stress impairs centromere function and prolongs SAC with increased temperatures. Centromere and its associated protein complex the kinetochores and the SAC contain a multitude of proteins, some of which are well characterized in *Arabidopsis thaliana*. Hence the authors could have used additional such tagged proteins to further strengthen their claim.

Indeed, several other proteins have recently been characterized as centromere/kinetochore components and could have been included in the study to further validate the results presented. To strengthen our argument, we have added new experimental data (Figure S4) showing temperature-induced depletion of CENH3 in wild-type plants by immunocytology. Thus, we convincingly show that temperature stress reduces the amount of CENH3. This is likely to affect the loading of most kinetochore and centromeric proteins. Here (new Figure S6) and in our previous study (Capitao et al., 2021), we have shown that genetic depletion of CENH3 in cenh3-4 mutants results in reduced loading of CENPC, MIS12 and BMF1 at mitotic centromeres and reduced loading of BMF3 and BMF1 at meiotic centromeres. We also attempted to assess the levels of CENPC and MIS12 on meiotic chromosomes by immunocytology, but our antibodies, which work on mitotic spreads, did not stain meiotic chromosomes.

Though the results presented here are interesting and solid, the study lacks a deeper mechanistic understanding of what causes the defective loading of CenH3 to the centromeres, and why the SAC protein BMF3 persists only at meiotic centromeres to prolong the spindle assembly checkpoint. Also, this observation should be interpreted in light of the fact that SAC is not that robust in plants as several null mutants of plant SAC components are known to grow as healthy as wild-type plants at normal growth conditions without any vegetative and reproductive defects.

Thank you for raising this point. We are of the opinion that SAC operates and it is important in plants - we have added a citation to a preprint from the Schnittger lab (Lampou et al., 2023, BioRxiv) that was published while this manuscript was under review. We think this is the most comprehensive analysis of plant SAC to date, clearly showing that SAC delays progression to anaphase in the presence of spindle inhibitors, although adaptation eventually occurs and the cell cycle progresses. This is very similar to the situation in animals, which also undergo spindle adaptation in similar situations. The difference between plants and animals may be due to subsequent events, where plants are better able to tolerate genome instability and resume cell division in the presence of abnormal chromosome numbers. Robustness and redundancy may be another reason why plant mutants deficient in SAC do not show obvious growth retardation.

One of the immediate responses to heat stress is the production of heat shock proteins(Hsps), which act as molecular chaperones to safeguard the proteome. It will be interesting to see if the expression levels of known HsPs can be correlated with their role in stabilizing the structure of SAC proteins like BMF1 to prolong its presence at the meiotic kinetochores.

Indeed, the heat stress response is likely to be involved in this process. We sought to investigate the role of this pathway by analyzing Arabidopsis mutants deficient in HEAT-SHOCK FACTOR BINDING PROTEIN (HSBP), which acts as a negative regulator of the heat shock response. This experiment was prompted by the observation that hsbp mutants have reduced fertility. We expected that an unrestricted heat stress response might affect meiosis and pollen formation. However, our initial experiments did not show altered pollen viability in response to heat stress in hsbp plants and we did not pursue this line of research further.

**Reviewer #3 (Public Review):**
Summary:Khaitova et al. report the formation of micronuclei during Arabidopsis meiosis under elevated temperatures. Micronuclei form when chromosomes are not correctly collected to the cellular poles in dividing cells. This happens when whole chromosomes or fragments are not properly attached to the kinetochore microtubules. The incidence of micronuclei formation is shown to increase at elevated temperatures in wild-type and more so in the weak centromere histone mutant cenH3-4. The number of micronuclei formed at high temperatures in the recombination mutant spo11 is like that in wild-type, indicating that the increased sensitivity of cenh3-4 is not related to the putative role of cenh3 in recombination. The abundance of CENH3-GFP at the centromere declines with higher temperature and correlates with a decline in spindle assembly checkpoint factor BMF1-GFP at the centromeres. The reduction in CENH3-GFP under heat is observed in meiocytes whereas CENH3-GFP abundance increases in the tapetum, suggesting there is a differential regulation of centromere loading in these two cell types. These observations are in line with previous reports on haploidization mutants and their hypersensitivity to heat stress.Strengths:This paper is an important contribution to our insights into the impact of heat stress on sexual reproduction in plants.Weaknesses:While it is highly significant, I struggled to interpret the results because of the poor quality of the figures and the videos.

We apologize for the poor quality of the figures. The figure resolution was drastically reduced during the conversion of the manuscript to pdf on publisher web site.

**Reviewer #1 (Recommendations For The Authors):**
To complete the presented analysis, it would be great to analyze the signal strength of the here-presented BMF3 reporter at high temps, see below for further reasoning.

Quantification of the BMF3 signal is difficult - it is only transiently associated with kinetochores and its level changes over time. Nevertheless, analysis of our movies taken under the same microscope settings indicates that the amount of BMF3 decreases with increasing temperature. This is illustrated in the new Figure S6C.

Conversely, how is the BMF1 and BMF3 signal strength in cenh3-4 mutants?

We performed an analysis of BMF1 and BMF3 signal in cenh3-4 mutants and observed a reduced level of signal from both proteins (Figure S6). In the case of BMF1, no signal was detectable in either somatic or meiotic cells.

How do the authors explain the reduction in BMF1 signal at 26 and 30{degree sign}C versus the extension of the duration of the SAC as measured by the persistence of a BMF3 signal (line 192: "...reduces the amount of CENH3 and the kinetochore protein BMF1 on meiotic centromeres, potentially affecting their functionality..." versus line 213: "...We observed that while the BMF3:GFP signal persisted, on average, for about 22.7 min at 21 and 26{degree sign}C, its appearance was prolonged to 40.5 min at 30{degree sign}C..."). Is the BMF3 signal also reduced at high temps (see question above)?

This is a very interesting point. While we see reduced levels of both proteins under heat stress or in cenh3-4 plants, the effect on BMF1 is much more pronounced and becomes undetectable under these conditions. This contrasts with BMF3, which appears to be reduced but is still clearly visible. These data suggest that BMF1 is more sensitive to reduced levels of CENH3 and it further corroborates the findings from the Schnittger lab that BMF1 is not the core component of SAC.

Line 18-20: The observation that heat stress reduces fertility has been made by several research teams before this study. I propose to write "confirm"/"support" etc. instead of "reveal" to avoid a (presumably not intended) false priority claim in the abstract.

We apologize, this was unintentional and we cite the relevant literature in the article. We have rewritten the abstract to avoid this impression.

Figure 2: The panel/legend appears to be a bit mixed up. Panel C is described in legend under A. In addition, I cannot find any blue arrows in panel A (which is described as panel B). Correspondingly, the references to the panels in this figure (lines 134/135 and following) need to be updated. I am also not sure how the meiocytes in this figure were stained. The dots look like centromeres but then their intensity rather increases with increasing temperature. If correct, how can this be reconciled with the authors' statement that centromeres decrease in size at higher temps?

We apologize for the mix up. An early version of the Figure was accidentally submitted and we now corrected it. The Panel B shows DAPI stained meiocytes at the tetrad stage and examples of micronuclei are indicated by arrowheads.

Line 520: Should read "genotype" not "phenotype".

Corrected

**Reviewer #2 (Recommendations For The Authors):**
(1) It is intriguing that heat stress impairs only the centromeres and segregation of meiotic chromosomes but not the mitotic chromosomes. No analysis of mitotic divisions is provided in the manuscript. As they have generated marker lines, it is reasonable to examine the mitotic time course as well by live monitoring of root tissues exposed to similar temperature conditions as done for meiotic analysis. This will help to address the effect of heat stress on mitotic centromeres and its comparison with meiosis will provide a better picture. There are two likely outcomes during mitosis:(a) It is possible that the heat stress also slows down mitotic progression as well as is the case in meiosis as shown in this paper and hence it is important to examine those as well to compare and contrast the CENH3/BMF1 dynamics in mitosis and meiosis.(b) The second scenario is that there is no effect of heat stress on the centromere integrity of mitotic chromosomes. In fact, the authors show indirect evidence in support of this wherein the eYFP: CENH3 showed a strong signal in the tapetal cells (somatic origin) surrounding the male meiocytes (generative origin). It is interesting that somatic cells of the tapetum show a strong signal whereas the meiocytes lack this. The authors should elaborate on this contrasting result.

The effect we observed seems to be specific to meiosis. We analyzed the progression of mitosis in root cells and we see a negligible effect of temperature on mitotic progression and no micronuclei formation. Interestingly, in terms of CENH3 loading, root cells show a slight decrease in CENH3 at 30°C, in contrast to the situation in tapetum cells. These and other data suggest a tissue/cell specific behavior of centromere maintenance and deserve further analysis. We plan to publish data on mitosis and tissue-specific aspects of CENH3 loading in a separate manuscript.

(2) Spindle assembly checkpoint (SAC) comprises several core proteins that are recruited to the kinetochores to correct the errors during the defective cell cycle. Here the authors demonstrate the prolonged presence of BMF3 as the only proof to claim that heat stress prolongs the spindle assembly checkpoint during metaphase I. Have the authors observed the dynamics of any other SAC core components such as MAD1, MAD2, MPS1, BUB3, and the like during heat stress?

No, we did not. We provide several independent lines of evidence that centromere structure and functionality are affected, and spindle checkpoint analysis is only one of them. At the time we designed these experiments, the only experimentally validated and well-characterized component of the SAC was BMF3, and we used only this protein as SAC reporter because a general analysis of the SAC was not the primary goal of our study. While this paper was under review, a preprint from the Schnittger lab focusing on plant SAC was published that comprehensively analyzed these SAC components in Arabidopsis and provided a solid foundation and resources for further research in this direction. This study also uses BMF3 as a reporter for SAC in meiotic cells. It is noteworthy that despite using different microscopic methods and different plant reporter lines, our labs independently arrived at exactly the same duration of BMF3 association with the kinetochore (i.e. 22 min).

(3) Is BMF1 a component of SAC or the kinetochore? I understand that BMF1 is a part of the core SAC ( Komaki and Schnittger, 2017) although it localizes to the kinetochore. There are well-characterized kinetochore proteins in Arabidopsis such as Mis12, NUF2, NNF1, and SPC24(MUN1) which the authors could have used as a kinetochore marker. Regardless, here the authors used it as a kinetochore marker. Being a part of SAC, one would expect the prolonged presence of BMF1 similar to BMF3 in the meiotic kinetochores but it is the other way. How to explain these contrasting results?

As discussed in the public section of the review, BMF1 does not seem to be the core component of SAC. Furthermore, this protein localizes to centromeres/kinetochore throughout the cell cycle and therefore, it cannot be used as SAC reporter.

(4) Micronuclei can form as a result of chromosome missegregation as shown for spo11-1 and also due to segregation error caused by DNA repair defects. Here it is not clear what is the origin of micronuclei. It is very hard to decipher from live cell imaging. A simple meiotic spread of anthers of different treatments would address the origin of micronuclei.

Cytology cannot easily determine the origin of micronuclei in meiotic cells. Acentric fragments produced from aberrant DNA repair will still be cytologically detectable only after metaphase I as they are tethered to the remaining chromatin via cohesion. Therefore, we took advantage of spo11 mutants that do not form any meiotic breaks, and hence cannot generate acentric fragments by aberrant repair, to discriminate the origin of micronuclei. We reason that all micronuclei produced in spo11 plants originate from chromosome mis-segregation and their increase at elevated temperature support the notion that heat stress further impairs chromosome segregation.

(5) Fig.1 B The microspores are not clearly visible in the alexander-stained anthers. It is not clear which is fertile and which is sterile. A better quality picture would be ideal to appreciate the fact.

Again, we apologize for poor quality of pictures due to manuscript conversion.

**Reviewer #3 (Recommendations For The Authors):**
(1) In Figure 2, it should be pointed out where the micronuclei are. I see here and there a single bright spot. In Arabidopsis, we have noticed bright spots under stress conditions that are autofluorescent signals. It needs to be shown that these spots are not observed in non-GFP lines. Better image quality may help too.

The micronuclei in Figure 2 are visualized by DAPI staining, not with GFP. The nuclei are now indicated by arrowheads.

(2) It was not possible to see the centromeres in Figure 3 hence I could not verify the fluorescence intensities of CENH3 and BMF1. There is also something wrong with the color codes blue and red in fig3B, C, and D.

Again, we apologize for poor quality of pictures due to manuscript conversion.

(3) Also in the videos it would help to point out where the micronuclei are seen. At what stage were these nuclei quantified? Given that meiosis progression in the cenh3-4 mutant is slower, it may be necessary to wait long enough to see established micronuclei. This information is supposed to be presented in Figure 2C. However, the X-axis shows time, not number. So I presume Fig 2C shows the duration of meiosis stages in the mutant. In Fig 2B, it shows the number of micronuclei per lobe. However, to correlate the incidence of micronuclei formation and the frequency of polyad formation (inviable microspores), one needs the quantification of the numbers of meiocytes carrying micronuclei. Then one can correlate the number of pollen per anther (shown in Fig 1c) with the incidence of micronuclei formation. The question of whether the degree of fertility reduction is due to micronuclei formation is a major issue that should be clarified.

Then micronuclei were not quantified from the movies, but from DAPI stained whole anthers at the tetrad stage as indicated in the main text. We also apologize for confusion with the Figure 2 as we mixed up the panels in the original submission. This has been corrected in the new submission.